# Investigation of Recognition and Classification of Forest Fires Based on Fusion Color and Textural Features of Images

**Cong Li \*, Qiang Liu, Binrui Li and Luying Liu**

School of Emergency Management and Safety Engineering, China University of Mining & Technology (Beijing), Beijing 100083, China
\* Correspondence: 18600156862@163.com; Tel.: +86-18600156862

**Abstract:** An image recognition and classification method based on fusion color and textural features was studied. Firstly, the suspected forest fire region was segmented via the fusion RGB-YCbCr color spaces. Then, 10 kinds of textural features were extracted by a local binary pattern (LBP) algorithm and 4 kinds of textural features were extracted by a gray-level co-occurrence matrix (GLCM) algorithm from the suspected fire region. In terms of its application, a database of the forest fire textural feature vector of three scenes was constructed, including forest images without fire, forest images with fire, and forest images with fire-like interference. The existence of forest fires can be recognized based on the database via a support vector machine (SVM). The results showed that the method's recognition rate for forest fires reached 93.15% and that it had a strong robustness with respect to distinguishing fire-like interference, which provides a more effective scheme for forest fire recognition.

**Keywords:** forest fire; Image recognition; color features; texture feature; gray level co-occurrence matrix

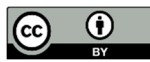

## 1. Introduction

The global area affected by forest fires is increasing yearly, and the demand for the rapid and efficient recognition of forest fires is gradually expanding. The frequent occurrence of forest fires in California has caused huge property losses, and tens of thousands of acres of land were burned in 2022. Forest fire recognition technology based on image features has significant advantages, such as high timeliness and a high recognition rate, which grants it the ability to identify forest fires as soon as possible to prevent their expansion in scale and replaces the traditional artificial lookout and artificial secondary image recognition methods that require high investments but yield poor results. It has been one of the main methods for forest fires' monitoring and identification and acts as an early warning solution.

With the continuous maturity of image acquisition and processing technology, scholars have thoroughly researched forest fires from the perspective of image recognition and have proposed a variety of detection and recognition methods. At the same time, the Handbook of Neural Computing [1] and the Handbook of Deep Learning Applications [2] introduced the extensive application of neural computing and deep learning, providing us with new ideas regarding forest fire image recognition. By setting hidden layer nodes directly, the ISSA (improved sparrow search algorithm) was used to input the corresponding weight and deviation as feature vectors for the rapid random configuration of flames in an FSCN (fast stochastic configuration network) grid and was trained using the extracted flame image and interference image feature vectors [3,4]. The method proposed by Roy et al. [5] is based on the LeNet5 convolutional neural network fire detection model combined with an L2-regularized non-sparse solution to classify the fire and non-

fire images in order to identify a fire in an outdoor environment, and it has achieved good measurement accuracy. Chen et al. [6] extracted multiple adjacent frames of flames, extracted dynamic features from the perspective of time and space, and described the process of the dynamic motion recognition of flame textures. The flames' structure and spatial features are extracted from the corresponding consistency information of the three-phase cross planes (the image sequence is divided into the three orthogonal directions), and the combined HOPC-TOP (which extracts PC features from spatial and temporal space) is used to identify the flame. Liu et al. [7] used an HOG (Histogram of Oriented Gradients) + an Adaboost classifier with a high recall rate to preliminarily identify possible forest fires, and then used a high-precision CNN (convolutional neural network) + an SVM (Support Vector Machine) classifier to further identify forest fire areas.

The recognition of forest fires from their color and textural features has achieved notable results. Zhang et al. [8] divided the core combustion region, used an Otsu–Kmeans flame image segmentation method to realize the regionalization segmentation of the flame target, extracted and input 10 feature vectors of the target region, constructed the model to output the corresponding combustion state, and then established an SVM vector machine for classification and recognition. Hosseini et al. [9] discussed a method employing deep learning to recognize flames and smoke, termed the "UFS-Net". The convolutional neural network structure is customized according to the flame for recognition. At the same time, a UFS data set (which includes a large number of images and videos collected from various data sources and artificial images for the training and evaluation of the UFS network) is used as the flame evaluation and training set, which is generally embodied as a computer vision-learning method. The UFS data set can also be used as the flame recognition training set. Wang et al.'s [10] convolutional neural network is often used for image feature learning. When combined with image processing, it can effectively and specifically learn to recognize flames and extract the corresponding features; furthermore, it has good performance and efficiency. Jiang et al. [11] applied a technique based on infrared images and flame spectrum threshold analysis to obtain a feature vector so as to quickly locate fires. Correspondingly, it can effectively eliminate various interferences in the forest and various noises in its own scene. In the research of Chen et al. [12], according to the temporal and spatial motion characteristics of flame, firstly, a Gaussian model was used to extract the flame's motion region; secondly, the flame recognition region was segmented through a flame-filtering algorithm; and finally, the recognition was made according to the statistics of the flame flicker frequency in the flame segmentation region. Muhammad et al. [13] proposed a new method for detecting forest fires using color and multi-color spatial local binary patterns based on flame and smoke characteristics and a single artificial neural network. It can detect various challenging flame and smoke regions. Hossain et al. [14] extracted the unique color and textural features of flames and formulated a variety of spatial color vector rules for flame segmentation to divide the feature region. However, due to the uncontrollable change in the brightness of a gray image, it is vulnerable to natural light and artificial light, which increases the error rate regarding flame recognition. Chen et al. [15] used flames' color space to filter local noisy feature points, and then used a SIFT (Scale Invariant Feature Transform) algorithm to extract fire feature values and converted them into feature vectors to identify fires. Kuang et al. [16] extracted the local textural features of a flame, reduced the dimension of the obtained feature vector via a principal component analysis algorithm, and substituted the obtained feature vector into the genetic algorithm for fire identification after an SVM calculation. Cui et al. [17] first used the watershed algorithm to extract the suspected flame area, and then selected four main flame characteristics as the recognition feature vector. Accordingly, the irregular sources are eliminated according to the irregular characteristics of the flame, which improves the degree of flame image processing and makes full use of eliminating interference sources and extracting flame characteristics. Prasad et al. [18] captured and processed the surface texture images and extracted 16 segmentation regions through preprocessing. The GLCM (Gray-level co-occurrence matrix) was used to extract and recognize the

feature quantity of the surface texture segmentation region, and the feature feedback was substituted into the SVM vector machine classifier. At the same time, it was also effectively applied in random forest texture recognition. Wang et al. [19] used a hog algorithm to extract image features from garbage. They put forward a new idea of an SVM to train the classification equipment and send relevant information to the database as recognized content. Liu et al. [20] proposed a flame detection algorithm based on saliency detection technology and a uniform local binary pattern that can reduce the false alarm rate of fire recognition technologies and improve the accuracy of sample classification. Ashour et al. [21] established an SVM vector machine classifier for processing the corresponding functions of different steps and judged its characteristics according to the drawn histogram. It also showed an excellent performance when it was substituted into the data set for machine learning.

At the same time, image recognition technology also plays an important role in the related research of fire, fire protection, and forest management. Liu et al. [22] proposed a new method of tree species identification and stock estimation for strengthening forest management. In this method, the forest images are collected by a digital camera. The method uses the UNET (the model used for the semantic segmentation of tree species images) network pre-trained by the VGG16 (as the encoder in the UNET network) model to accurately identify the number of trees and tree species contained in the image. In order to advance the use of UAVs in forest measurement, Seifert et al. [23] chose to use video clips obtained from flight at multiple altitudes in combination with commercial multi-view reconstruction software and a multivariate, generalized additive model for analysis in order to set the best flight parameters and select sensor resolution. Lai et al. [24] combined flames' surface, invisible heat flow, and temperature into an image recognition system. By changing the forced airflow size and wind direction of the micro wind tunnel, the combustion intensity was studied, the flame combustion and propagation process were identified, and the flame's temperature and material surface temperature were monitored. He et al. [25] used the image recognition method to quantitatively study the influence of a tunnel's longitudinal ventilation speed on the intermittent combustion behavior and flame injection behavior of a car. Zhao et al. [26] studied the combustion behavior of a floating roof tank in a chemical industry park. This paper introduces an image recognition method based on images of the flame's profile, which is used to analyze the necking and periodic fluctuation of flame under different diameter oil pans and different buoyancy plume conditions. Zhao et al. [27] used the RGB (Red Green Blue) color rule to determine a flame's shape through the difference between the flame and the background, analyzed the flame diffusion and combustion behavior, and explored the influence of slope on the flames' spread and height with respect to the leaked oil. Li et al. [28] captured the flame combustion signal through high-speed photography to determine the diffusion combustion behavior in a tube, which is manifested as the influence on the ignition mechanism and the propagation of the Flame Shock Wave in the tube under pressure change conditions.

Previous studies on the characteristics of forest fire images focused on images' color, texture, and motion detection. However, the influence of fire-like interference sources (red objects that interfere with fire image recognition in forests, such as banners, maple leaves, bottles, etc.) on forest fire recognition has not been adequately considered, and the recognition accuracy still needs to be further improved. In this paper, a forest fire recognition method based on fusion color and textural features was investigated. The suspected forest fire region was detected and segmented via the fusion RGB-YCbCr (Y is the luminance component of the color, while CB and CR are the concentration offset components of blue and red) color spaces. A 14-dimensional vector of the forest fire image was formed by LBP (Local Binary Patterns) and a GLCM (gray-level co-occurrence matrix) algorithm. Consequently, a forest fire can be recognized via comparison to the database of the vectors through the SVM, and the method was verified to have a high accuracy for forest fire recognition.

## 2. Method

The technological roadmap of the method is presented in Figure 1. There are four main steps to achieve accurate recognition. Firstly, preprocessed images and the suspected flame area were extracted based on the fused RGB-YCbCr color spaces. Secondly, the LBP and GLCM algorithms were used to extract the textural features of the suspected flame area, and a 14-dimensional texture feature vector was formed. Thirdly, the database of textural features was established based on a large number of training images, including normal forest images, forest fire images, and fire-like interference images. Fourthly, forest fires could be identified by judging the images' similarity with three types of images in the database via support vector machine. In addition, the accuracy of the method was further improved by expanding the forest fire image database continuously.

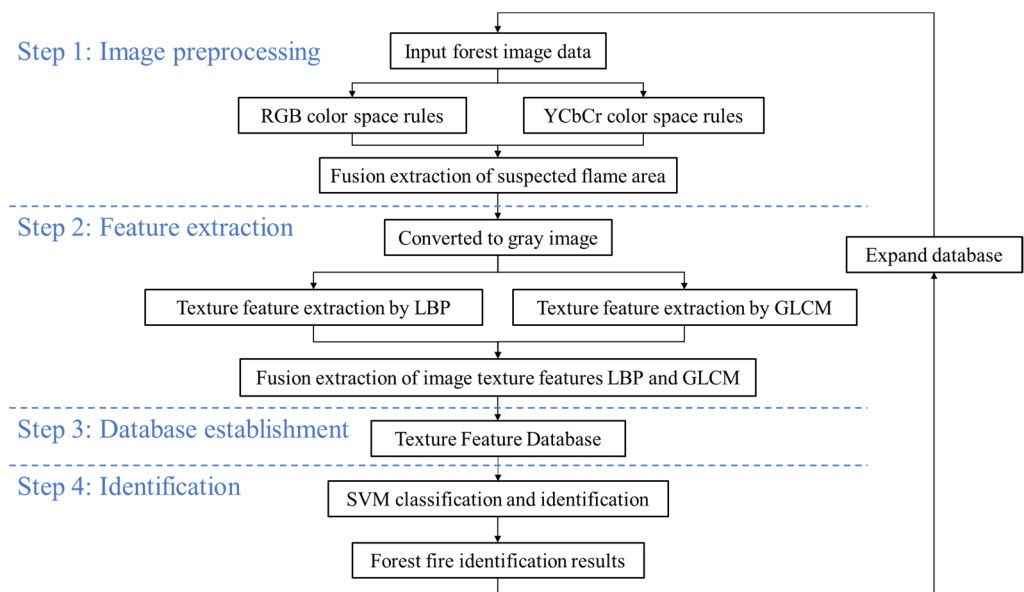

**Figure 1.** Forest Fire Identification Process.

### 2.1. Segment of Suspected Flame Region

### 2.1.1. Segment via RGB Color Space

RGB mode is the kind of color of a mature system; most displays have adopted the RGB mode . The $R$ part of the flame pixel is larger than the $G$ part and the $B$ part; the difference between $R$, $G$, and $B$ can be used to extract the red suspected flame pixel in the forest image. RGB color rules are as follows:

$$R_{I_{(x,y)}} = \begin{cases} I_{(x,y)}, R_{(x,y)} - G_{(x,y)} > R_{GT}, R_{(x,y)} - B_{(x,y)} > R_{BT} \\ 0, else \end{cases} \quad (1)$$

Here, $R_{GT}$ is the red–green color threshold and $R_{BT}$ is the red–blue color threshold. Select 30, 40, 50, 60, and 70 for $R_{GT}$ and $R_{BT}$, and calculate the extraction results and pixel retention rate of RGB algorithm under different $R_{GT}$ and $R_{BT}$ conditions [29–31]. Our results are shown in Figures 2 and 3:

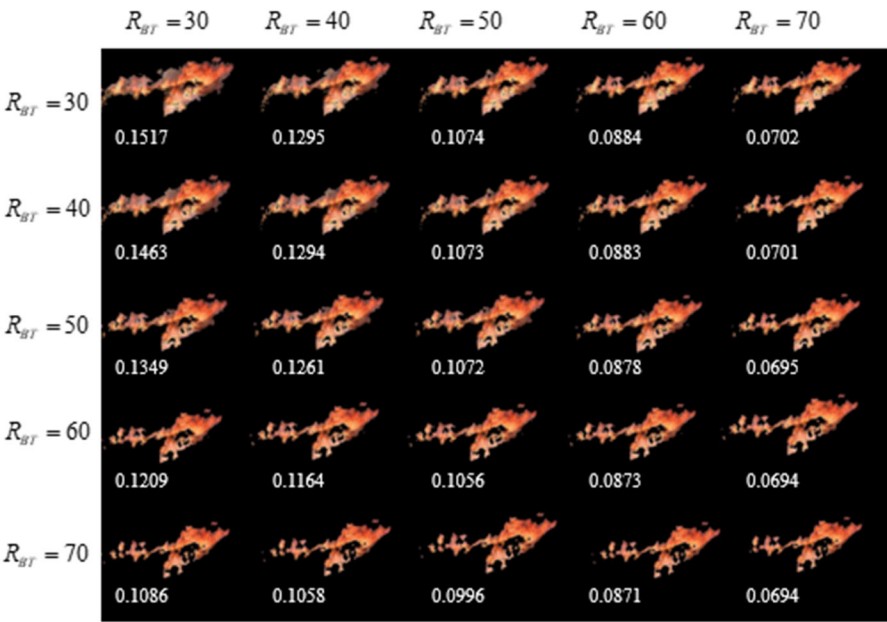

**Figure 2.** Pixel retention rate of different $R_{GT}$ and $R_{BT}$ (thresholds).

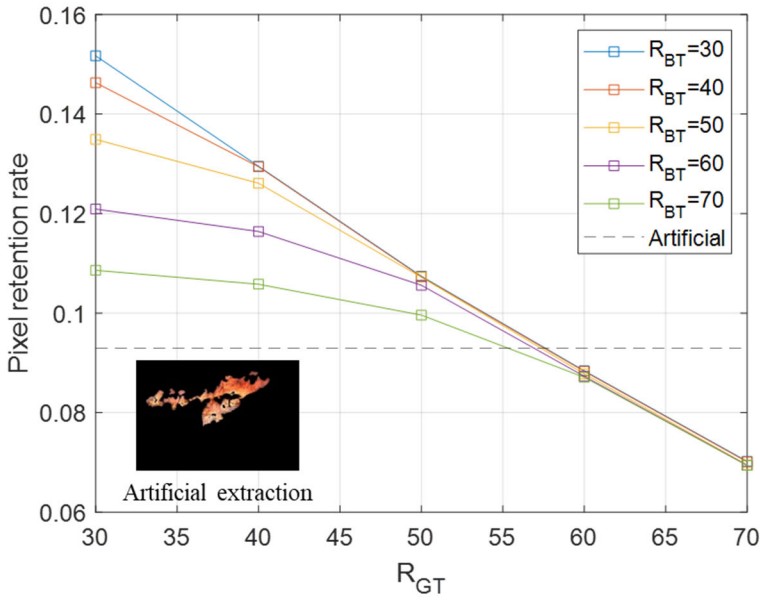

**Figure 3.** The relationship between $R_{GT}$, $R_{BT}$, and pixel retention rate.

Figure 3 shows that $R_{GT}$ and $R_{BT}$ are inversely proportional to the extracted reserved pixel rate. Since the green concentration of trees and the red concentration of flames in forest images are large, the difference between $R$ and $G$ can better reflect the difference between flame pixels and forest pixels; so, $R_{GT}$ is the main index for extracting flame components. When $R_{GT}$ is large, the extraction result is mainly affected by $R_{GT}$ value and is relatively less affected by $R_{BT}$ value. Comparing the extraction results of different $R_{GT}$ and $R_{BT}$ with the manual extraction results yields the accuracy of flame extraction, as shown in Table 1.

**Table 1.** Extraction accuracy of RGB color space with different values.

| Threshold | | $R_{GT}$ | | | | |
|---|---|---|---|---|---|---|
| | | **30** | **40** | **50** | **60** | **70** |
| $R_{BT}$ | 30 | 0.6249 | 0.6534 | 0.7126 | 0.7743 | 0.7683 |
| | 40 | 0.7095 | 0.7097 | 0.7255 | 0.8051 | 0.7957 |
| | 50 | 0.7545 | 0.7545 | 0.7552 | 0.7584 | 0.7695 |
| | 60 | 0.7200 | 0.7200 | 0.7200 | 0.7201 | 0.7186 |
| | 70 | 0.6415 | 0.6416 | 0.6416 | 0.6416 | 0.6417 |

Table 1 shows that when $R_{GT}$ = 60 and $R_{BT}$ = 40, the extraction effect is the best, as it can accurately exclude non-flame pixels and preserve the burning area of the forest. Some extraction results via the RGB color space of the forest image without fire, with fire, and with fire-like interference are shown in Figure 4. It can be seen that the deficiency of the RGB color space is that some red fire-like interference may be misjudged as fire in Figure 4c.

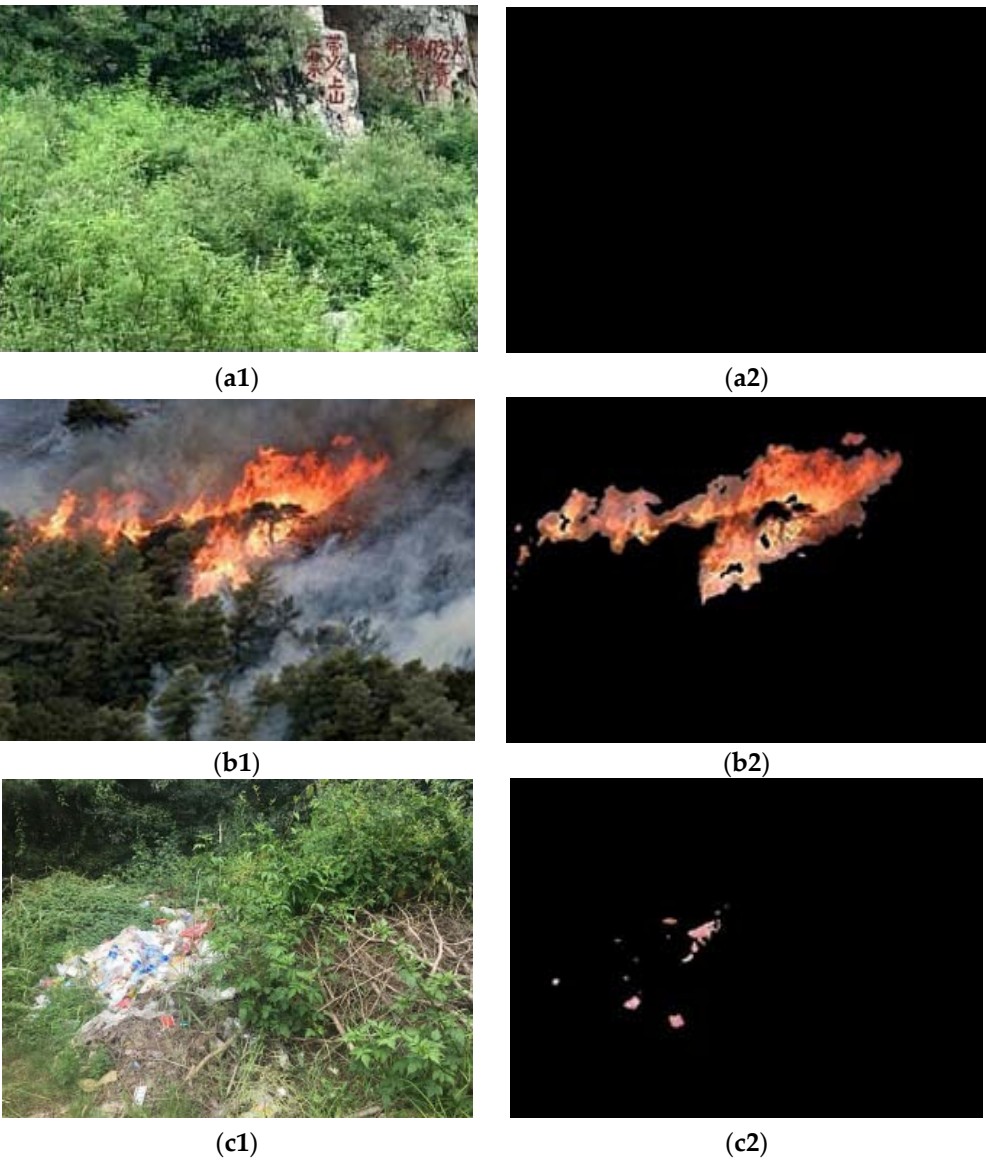

(**a1**)  (**a2**)

(**b1**)  (**b2**)

(**c1**)  (**c2**)

**Figure 4.** Extraction results via RGB color space: (**a1**) Forest image without fire; (**a2**) extraction results of a forest image without fire; (**b1**) forest image with fire; (**b2**) extraction results of forest image with fire; (**c1**) forest image with fire-like interference; (**c2**) extraction results of forest image with fire-like interference.

2.1.2. Segment via YCbCr Color Space

YCbCr color space mainly focuses on image brightness features, and it can extract suspected flame pixels with high brightness in the image. Conversion formula from RGB space to YCbCr space is as follows: RGB (0~255)

$$\begin{bmatrix} Y \\ C_b \\ C_r \end{bmatrix} = \frac{1}{256} \times \begin{bmatrix} 0.2568 & 0.5041 & 0.0979 \\ -0.1482 & -0.2910 & 0.4392 \\ 0.4392 & -0.3678 & -0.0714 \end{bmatrix} \times \begin{bmatrix} R \\ G \\ B \end{bmatrix} + \begin{bmatrix} 16 \\ 128 \\ 128 \end{bmatrix} \quad (2)$$

YCbCr color rules are as follows:

$$R_{II(x,y)} = \begin{cases} I_{(x,y)}, Y_{(x,y)} > Y_{mean}, C_{b(x,y)} < C_{bmean}, C_{r(x,y)} > C_{rmean} \\ 0, else \end{cases} \quad (3)$$

Here, $Y_{mean}$ is the mean value of the brightness of the original image, $C_{rmean}$ is the mean value of the red concentration component of the original image, and $C_{bmean}$ is the mean value of the blue concentration component of the original image [32]. The extraction results are shown in Figure 5. The deficiency of the YCbCr color space is that some green plants may be recognized as fire, as in Figure 5a,c.

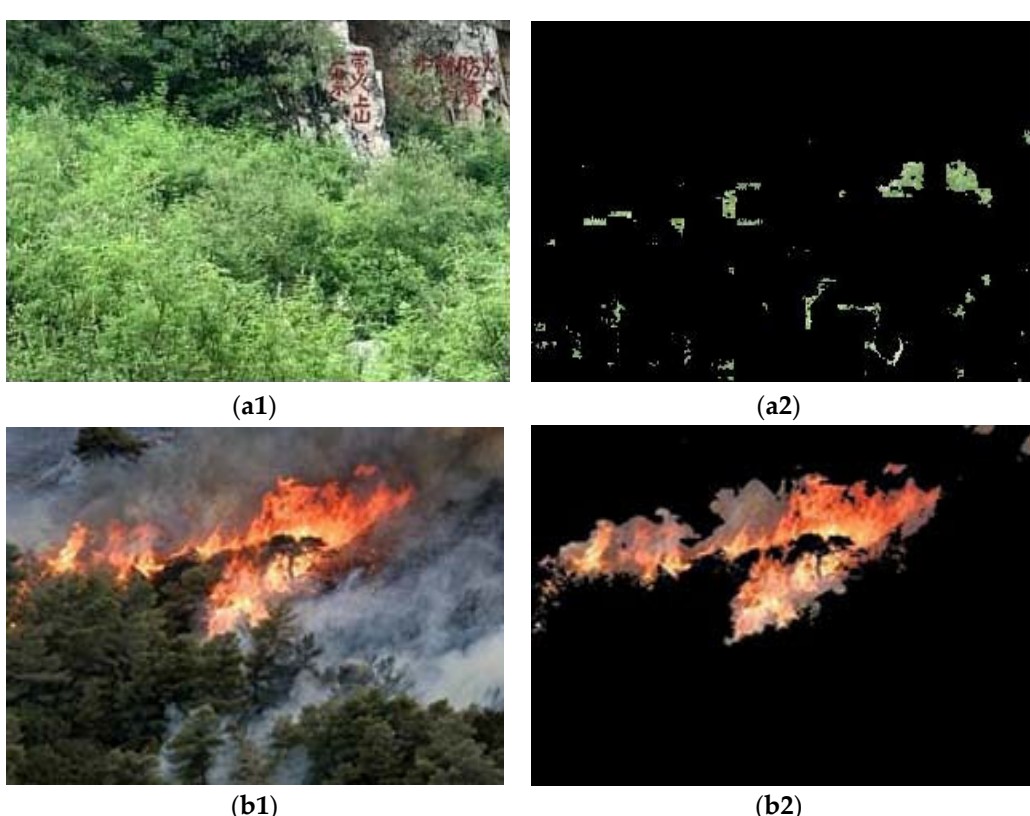

(a1)　　　　　　　　(a2)

(b1)　　　　　　　　(b2)

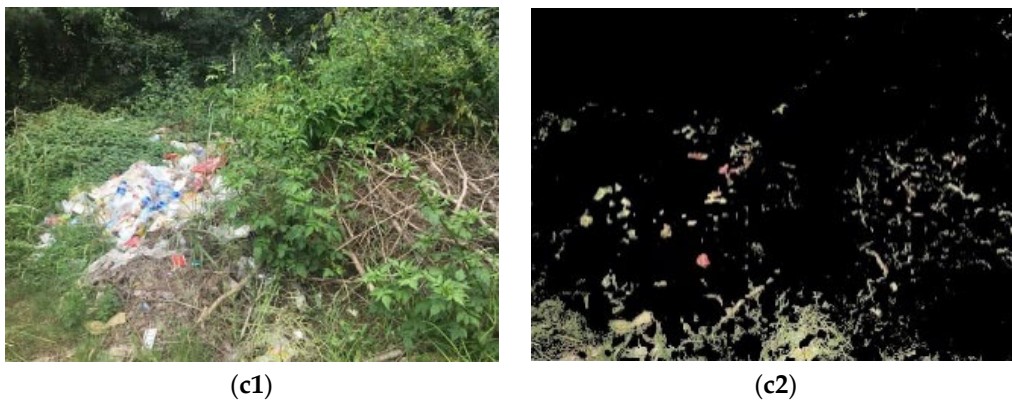

(c1)          (c2)

**Figure 5.** Extraction results via YCbCr color space: (**a1**) Forest image without fire; (**a2**) extraction results of forest image without fire; (**b1**) forest image with fire; (**b2**) extraction results of forest image with fire; (**c1**) forest image with fire-like interference; (**c2**) extraction results of forest image with fire-like interference.

### 2.1.3. Segment via Fusion RGB-YCbCr Color Spaces

Comparing the extraction results of the RGB and YCbCr color spaces of the forest fire images, the extraction of RGB color space was more accurate for the region with large value of the red component, while it could not accurately exclude some low-brightness interferences. However, the YCbCr color space was more accurate in extracting high-brightness flames, but some non-red pixels could be included. Combined with the advantages of the two kinds of color spaces, and by determining the intersection of the two results, the fusion of the RGB and YCbCr color spaces was applied to extract the suspected fire region to improve the accuracy of the segment [33]. The comprehensive extraction rule is written as follows:

$$R_{\mathrm{III}(x,y)} = \begin{cases} I_{(x,y)}, R_{\mathrm{I}(x,y)} \neq 0, & R_{\mathrm{II}(x,y)} \neq 0 \\ 0, & \text{else} \end{cases} \tag{4}$$

In addition, the accuracy of the segment was obtained by comparing different algorithms with manual extraction results. When RGB and YCbCr color models are used alone, the accuracy of segment was 0.8051 and 0.6522, respectively. However, when the fused RGB-YCbCr color spaces are used, the accuracy of segment can be raised to 0.8568, which is better than the traditional RGB or YCbCr color models. The comparison of RGB, YCbCr, and fusion RGB-YCbCr is shown in Figure 6.

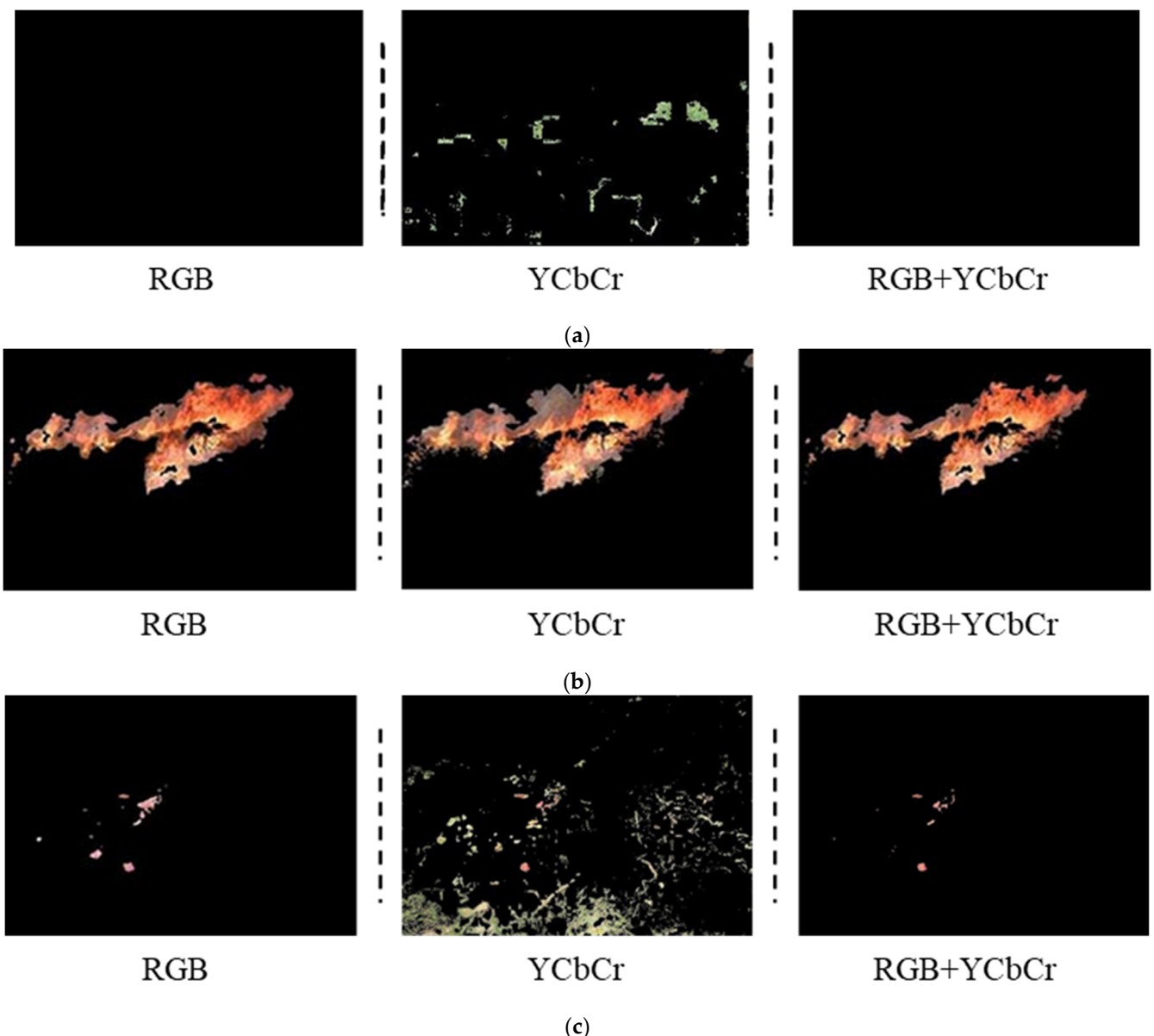

**Figure 6.** Comparison of RGB, YCbCr, and fusion RGB-YCbCr results: (**a**) extraction results of forest image without fire; (**b**) extraction results of forest image with fire; (**c**) extraction results of forest image with fire-like interference.

Figure 6 indicates that by comparing the different segment results with the original image, some green and red interferences were excluded, and the fire area was successfully extracted, indicating that the algorithm has strong robustness. From the comparison between Figure 6b,c, there were obvious differences between the textural features of flame and interference sources. Therefore, considering the special textural features of forest fires, the LBP and GLCM algorithm are used to characterize the textural feature-based information of a suspected fire region for recognition and classification.

### 2.2. Extraction of Textural Features

The textural features of forest fire images generally have a sheet or plane distribution, with dense texture and strong continuity in the central area. The textural features of sunsets are mainly stripes and not densely distributed. The textural features of red leaf forests generally have a point distribution, and the continuity of their central region is poor . The textural features of red stripes are generally continuous and concentrated. The difference

in the image textures between forest fire flames and a fire-like interference source can be used to classify and recognize flames in forest fire images.

### 2.2.1. Extraction of textural Features via LBP

LBP algorithm can be used to describe the textural features of forest fire images [34]. With the continuous development and application of the algorithm, the basic LBP algorithm has become too fixed [35]. Ojala et al. [36] extended the LBP algorithm and proposed a uniform pattern, rotation-invariant pattern, and rotation-invariant uniform pattern. The uniform pattern meets the requirement of reducing the number of feature vectors for LBP algorithm. The calculation method is as follows:

$$U(LBP_{P,R}) = \sum_{i=0}^{P-1} \left| s(g_{(i+1)\bmod P}) - s(g_i - g_c) \right| \tag{5}$$

where $g_c$ is the Gray value of the central pixel. $g_i$ is the Gray value of neighborhood pixels. $P$ denotes the number of pixels around. $R$ represents the neighborhood radius, which is the Euclidean distance between the central pixel and the neighborhood pixels. Rotation invariant pattern can solve the problem wherein the $LBP$ value changes due to image rotation or tilt, and thus keeps $LBP$ value unchanged. The calculation method is as follows:

$$LBP_{P,R}^{ri} = \min_{0 \le i \le P-1} \left\{ ROR\left(LBP_{P,R}, i\right) \right\} \tag{6}$$

$ROR(x, j)$ performs a rotation operation that moves the $x$ loop to the right by $i$ bits. Rotation invariant uniform pattern is obtained by combining rotation-invariant pattern with uniform mode. The calculation method is as follows:

$$LBP_{P,R}^{riu2} = \begin{cases} \sum_{i=0}^{P-1} s(g_i - g_c), U(LBP_{P,R}) \le 2 \\ P+1, else \end{cases} \tag{7}$$

This experiment mainly studies the $LBP_{(8,1)}^{riu2}$ algorithm and $LBP_{(8,2)}^{riu2}$ algorithm with fewer feature vectors. Taking the $LBP_{(8,2)}^{riu2}$ algorithm as an example, Figure 7 shows the LBP texture of various forest fire images, and Figure 8 shows the specific values of the three scenes. In other words, there are 10 values of textural features extracted from the forest image via the $LBP_{(8,2)}^{riu2}$ algorithm, which can achieve the recognition and classification of objects in the following section.

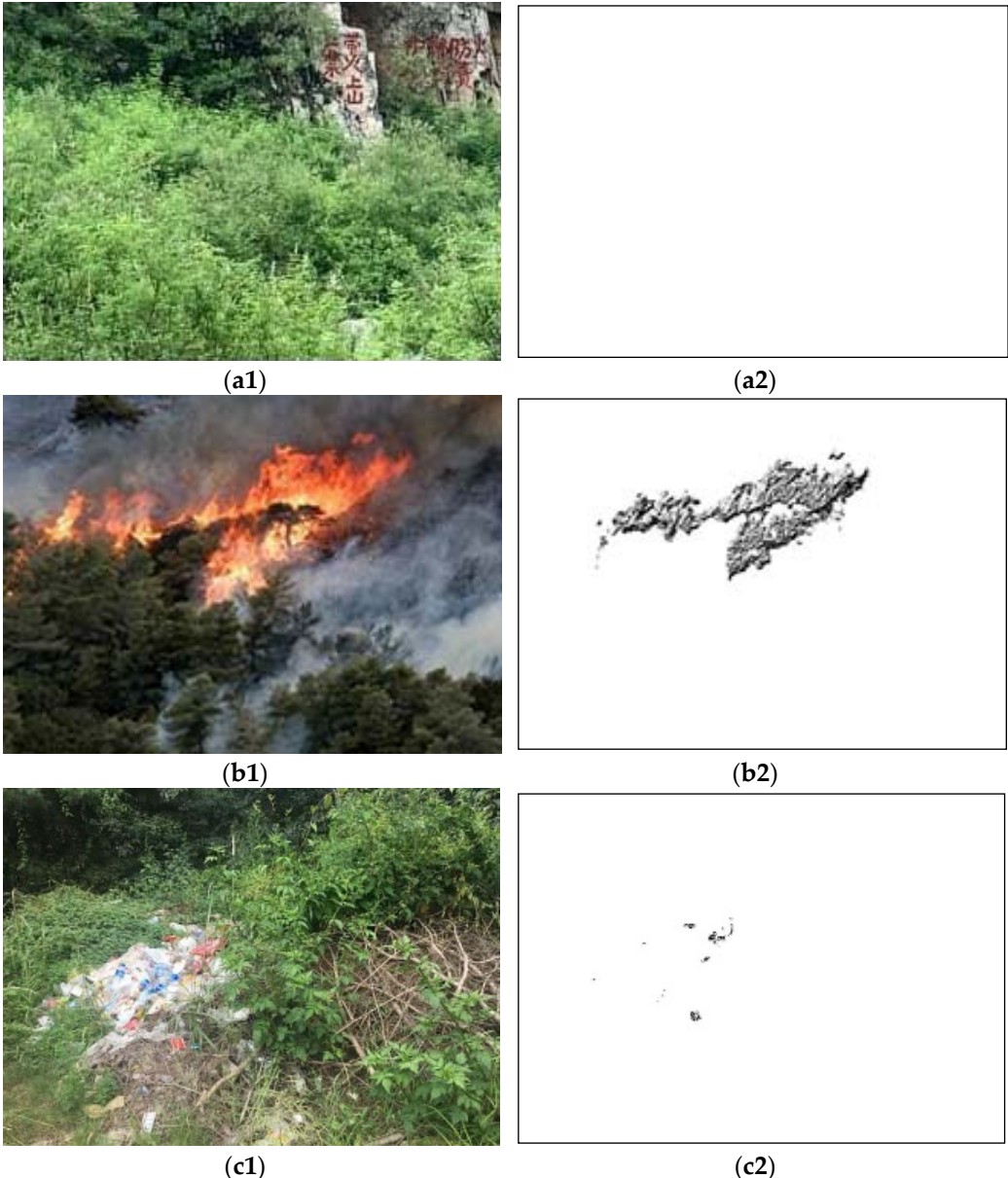

**Figure 7.** LBP-extracted image textures of forest fire images: (**a1**) forest image without fire; (**a2**) LBP-extracted image textures of forest image without fire; (**b1**) forest image with fire; (**b2**) LBP-extracted image textures of forest image with fire; (**c1**) forest image with fire-like interference; (**c2**) LBP-extracted image textures of forest image with fire-like interference.

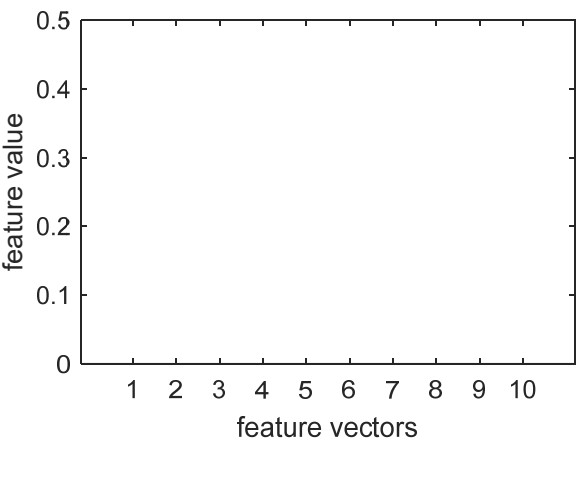

(a)

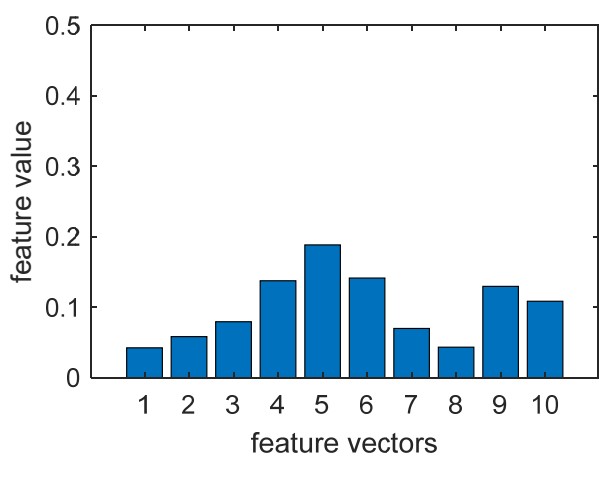

(b)

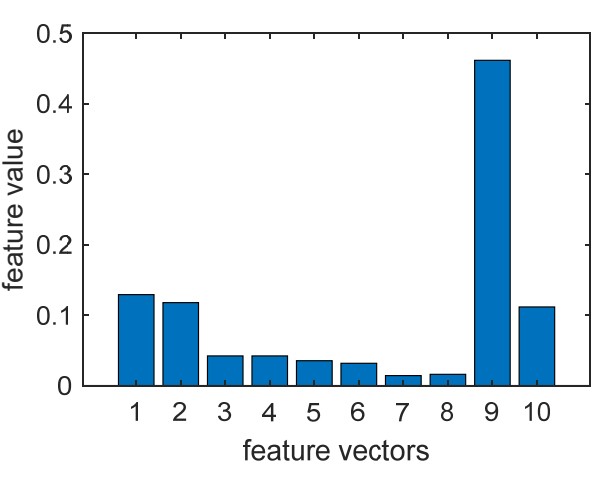

(c)

**Figure 8.** Histograms of various types of forest fire images: (**a**) $LBP_{(8,2)}^{riu2}$ histograms of forest image without fire; (**b**) $LBP_{(8,2)}^{riu2}$ histograms of forest image with fire; (**c**) $LBP_{(8,2)}^{riu2}$ histograms of forest image with fire-like interference.

### 2.2.2. Extraction of Textural Features via GLCM

Gray level co-occurrence matrix is an algorithm that obtains the textural features of the images by counting the gray levels of two pixels at a relative position in the image. [37–39]. The commonly used statistical features of GLCM algorithm include angular second moment, contrast, inverse different moment, and correlation.

Angular second moment is the sum of the squares of the values of the gray level co-occurrence matrix, which represents the thickness of a fire's texture and the uniformity of the gray distribution, as in Equation (8):

$$A = \sum_{b_1} \sum_{b_2} \left( C_{b_1,b_2} \right)^2 \tag{8}$$

Contrast is the relationship between a pixel value and an adjacent pixel value, which measures the depth and clarity of forest fire image textures, as in Equation (9):

$$C_1 = \sum_{b_1}\sum_{b_2}(b_1 - b_2)^2 C_{b_1,b_2} \tag{9}$$

The inverse different moment measures the degree of textural change and smoothness of the local area of a forest fire image, as in Equation (10):

$$I = \sum_{b_1}\sum_{b_2}\frac{C_{b_1,b_2}}{1+(b_1-b_2)^2} \tag{10}$$

Correlation measures the similarity of spatial gray level co-occurrence matrix elements in row or columnal directions and indicates the linear relationship of gray forest fire gray image, as Equation (11):

$$C_2 = -\sum_{b_1}\sum_{b_2}\frac{(b_1 - \mu_{b_1})(b_2 - \mu_{b_2})C_{b_1,b_2}}{\sigma_{b_1}\sigma_{b_2}} \tag{11}$$

Here, $m_x$ is the sum of each column element in matrix $C_2$, $m_y$ is the sum of each row element in matrix $C_2$, and $\mu_{b_1}$, $\mu_{b_2}$, $\sigma_{b_1}$, and $\sigma_{b_2}$ are the mean and standard deviation of $m_x$ and $m_y$. Figure 9 presents the 4 eigenvalues of the three scenes extracted via GLCM algorithm, and the feature vectors are shown in Table 2. It can be seen that the GLCM eigenvalues of the image without fire are basically zero, while those of the image with fire-like interference present a larger contrast and a negative correlation. The trend of these eigenvalues can be used as a criterion for recognition and classification.

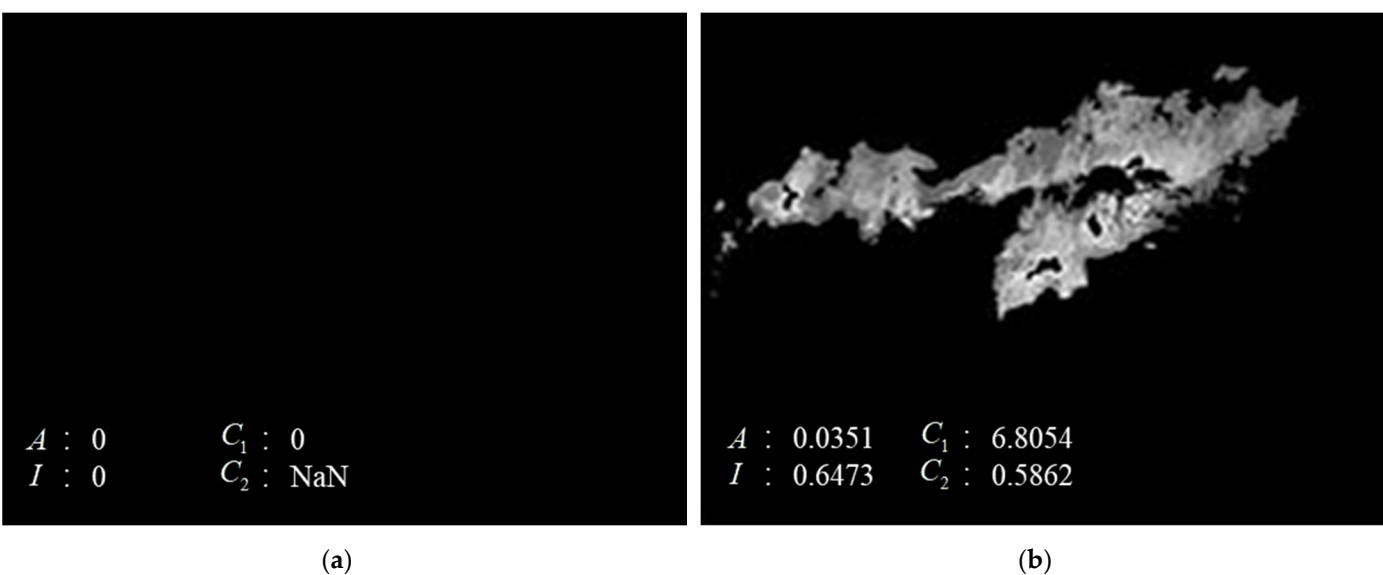

(**a**)                                               (**b**)

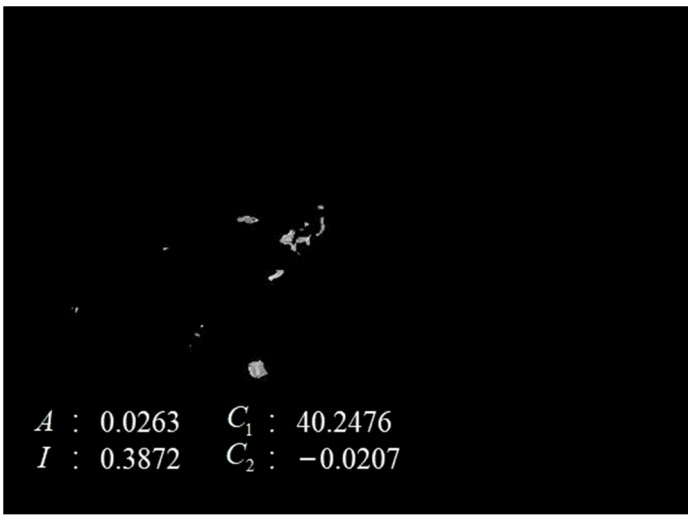

(**c**)

**Figure 9.** Gray scale image**:** (**a**) forest image without fire; (**b**) forest image with fire; (**c**) forest image with fire-like interference.

**Table 2.** Gray level co-occurrence matrix feature vector.

| Picture Types | $A$ | $C_1$ | $I$ | $C_2$ |
|---|---|---|---|---|
| Forest image without fire | 0 | 0 | 0 | NaN |
| Forest image with fire | 0.0351 | 6.8054 | 0.6473 | 0.5862 |
| Forest image with fire-like interference | 0.0263 | 40.2476 | 0.3872 | −0.0207 |

By analyzing and comparing the LBP feature and GLCM feature extraction results of forest images with fire and forest images with fire-like interference sources, a significant difference was discovered between the two. When analyzing the fire's textural features, the combination of the two can complement each other and improve the accuracy of recognizing flames in forest fire images. The feature vectors extracted by the two were combined to form a new 14-dimensional feature vector to describe the textural features of forest fire flames [40].

*2.3. Classifier*

In this section, the support vector machine was used to construct a decision function that recognizes and classifies the forest images in the three scenes. The purpose of SVM algorithm is to construct a decision function that can classify data to the greatest extent. All sample data correspond to the following formula:

$$\underset{w,b,\xi}{Minimize}\frac{1}{2}\langle w \cdot w\rangle + C\sum_{i=1}^{n}\xi_i \tag{12}$$

where $w$ is the normal vector of the hyperplane, $b$ is the intercept of the hyperplane, and $C$ is the penalty parameter. The LBP histogram distribution feature and GLCM feature of a forest fire image with flames and an interference image are extracted from the existing samples to form 14-dimensional vector: X = [$L_1$, $L_2$, $L_3$, ..., $L_{10}$, $A$, $C_1$, $I$, $C_2$]. The collected data are used to establish training set and test set, and Radial Basis Function (RBF) kernel is used to identify and classify forest fire flames [41–43]. RBF kernel is defined as follows:

$$K\left(x_i, x\right) = e^{-\frac{\|x-x_i\|^2}{2p^2}} \tag{13}$$

where $p$ is the width of the radial basis kernel function. Using the radial basis function kernel to identify forest fires is effective. RBF can analyze high-dimensional functions, for which the main identification classification steps are as follows.

(1) Training process: The target image is extracted as the training set, based on the LBP histogram feature of the target image and the GLCM texture extraction feature as the image feature input to the SVM vector machine for classifier training.

(2) Recognition process: The LBP and GLCM features of the recognition image are extracted and classified by the trained classifier. Finally, the classification performance of the classifier is evaluated by the accurate recognition results.

This paper is based on a support vector machine used to identify and classify forest fires and combines comparative convolutional neural networks to verify the accuracy and effectiveness of the algorithm. Convolutional neural networks are a kind of feedforward neural network employing convolution calculations and possessing a deep structure, and it is one of the representative algorithms of deep learning. Convolutional neural network has mature applications in the field of computer vision, and this paper uses its image recognition capability to recognize the fire in forest fire images. First, this algorithm uses the fusion color space rule to extract the suspected flame area, then uses the LBP-GLCM method to extract the textural features, and finally inputs the textural features into the support vector machine. As a comparison algorithm, we input the extracted image of the suspected flame area into the convolutional neural network.

### 3. Results and Discussion

This method was implemented in MATLAB R2019a. The operating system of the experimental platform was Windows 11, and the processor was Intel® Core™ i5—9300H. A total of 1317 forest images were collected from the field, including 513 forest images without fire, 420 forest images with fire, and 384 forest images with fire-like interference. The forest images included a variety of common trees in China, such as birch, pine, cypress, etc. The forest fire images selected were mainly close range or high-definition images because their color and textural features are more obvious. The types of fire-like interference included red garbage, red stripes, a red leaf forest, etc. Some of the images of the recognition process regarding the three scenes are presented as Figure 10. Figure 10a is a normal forest image without fire, and obviously there was no extracted fire region after the color extraction process. Figure 10b is a forest image with fire, while Figure 10c is a red banner in the forest, which was used as interference in this study.

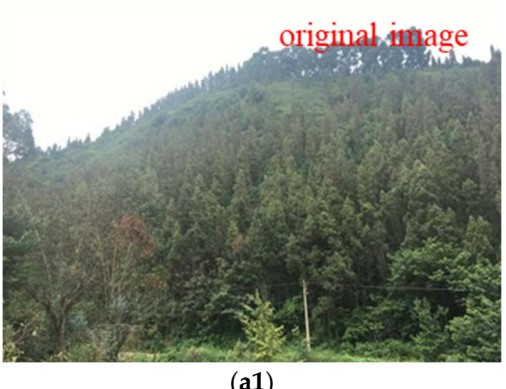
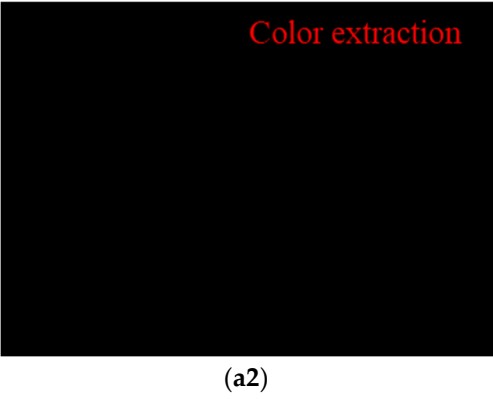

(**a1**)　　　　　　　　　　　　　　　　(**a2**)

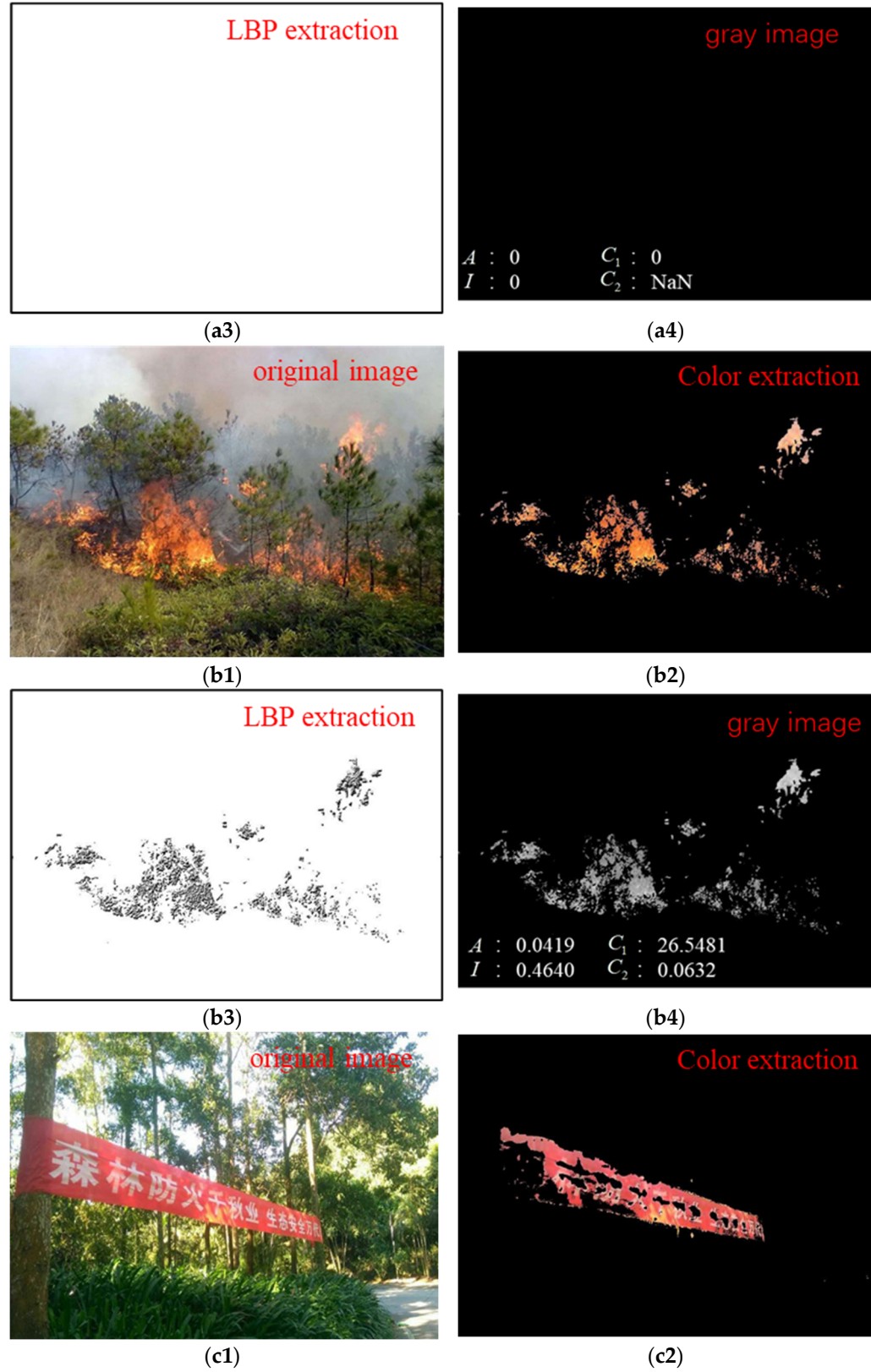

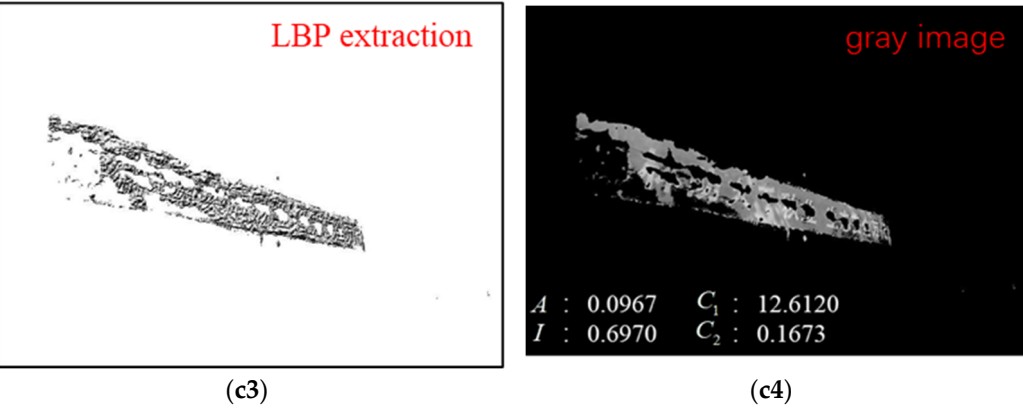

(**c3**)                                                                                                                                            (**c4**)

**Figure 10.** Experimental image sample and extraction results: (**a1**) forest image without fire; (**a2**) color extraction result of forest image without fire; (**a3**) LBP extraction result of forest image without fire; (**a4**) gray image of forest image without fire; (**b1**) forest image with fire; (**b2**) color extraction result of forest image with fire; (**b3**) LBP extraction result of forest image with fire; (**b4**) gray image of forest image with fire; (**c1**) forest image with fire-like interference; (**c2**) color extraction result of forest image with fire-like interference; (**c3**) LBP extraction result of forest image with fire-like interference; (**c4**) gray image of forest image with fire-like interference.

Table 3 shows the image and feature vectors of the typical image samples used in this study, and Table 4 shows the forest fire identification results under different algorithms.

**Table 3.** Fusion feature vector of partial forest sample image and color texture.

|  | Forest without Fire | Forest Image with Fire 1 | Forest Image with Fire 2 | forest Image with Fire 3 | Interference 1 | Interference 2 | Interference 3 |
|---|---|---|---|---|---|---|---|
|  |  |  |  |  |  |  |  |
| $L_1$ | 0.0000 | 0.1187 | 0.1292 | 0.0976 | 0.0942 | 0.1417 | 0.1196 |
| $L_2$ | 0.0000 | 0.1021 | 0.1071 | 0.0922 | 0.1077 | 0.0560 | 0.1129 |
| $L_3$ | 0.0000 | 0.0538 | 0.0530 | 0.0759 | 0.0676 | 0.0101 | 0.0509 |
| $L_4$ | 0.0000 | 0.0472 | 0.0472 | 0.0806 | 0.0697 | 0.0126 | 0.0480 |
| $L_5$ | 0.0000 | 0.0413 | 0.0369 | 0.0697 | 0.0601 | 0.0062 | 0.0339 |
| $L_6$ | 0.0000 | 0.0300 | 0.0308 | 0.0532 | 0.0268 | 0.0022 | 0.0170 |
| $L_7$ | 0.0000 | 0.0180 | 0.0266 | 0.0313 | 0.0106 | 0.0000 | 0.0089 |
| $L_8$ | 0.0000 | 0.0206 | 0.0361 | 0.0282 | 0.0047 | 0.0000 | 0.0074 |
| $L_9$ | 0.0000 | 0.4034 | 0.3329 | 0.3245 | 0.4617 | 0.7557 | 0.5004 |
| $L_{10}$ | 0.0000 | 0.1647 | 0.2002 | 0.1469 | 0.0967 | 0.0155 | 0.1011 |
| $A$ | 0.0000 | 0.0019 | 0.0019 | 0.0022 | 0.0018 | 0.0031 | 0.0011 |
| $C_1$ | 0.0000 | 0.9776 | 0.9471 | 0.9424 | 0.9782 | 1.0018 | 0.9925 |
| $I$ | 0.0000 | 0.0188 | 0.0296 | 0.0336 | 0.0184 | 0.0042 | 0.0098 |
| $C_2$ | NaN | 0.0017 | 0.0214 | 0.0217 | 0.0016 | −0.0091 | −0.0033 |

Table 3 shows that the difference in the textural feature vector between different images is large. In particular, the first 13 terms of the textural feature vector of the forest image without fire are all 0. We can use the different characteristics of the textural features of the different types of images to identify forest fires and input the textural information of the three types of images as feature vectors into the support vector machine.

**Table 4.** Forest fire flame identification results.

| Algorithm | Vector Dimensions | Sample Size | Number of Correct Recognitions | Accuracy |
|---|---|---|---|---|
| $LBP_{(8,1)}^{riu2}$ | 10 | 190 | 156 | 82.11 |
| $LBP_{(8,2)}^{riu2}$ | 10 | 190 | 163 | 85.78 |
| $GLCM$ | 4 | 190 | 161 | 84.73 |
| $LBP_{(8,1)}^{riu2} + GLCM$ | 14 | 190 | 174 | 91.58 |
| $LBP_{(8,2)}^{riu2} + GLCM$ | 14 | 190 | 177 | 93.15 |

The results in the table indicate that the recognition rate concerning forest fire flame images was low when the LBP or GLCM algorithms were used alone. The two algorithms were fused, and the 14-dimensional fusion feature vector was obtained. By using the SVM classifier, the recognition rate for forest fires can reach more than 90%, and the fusion algorithm $LBP_{(8,2)}^{riu2} + GLCM$ can reach 93.15%, wherein both applications can accurately identify forest fires. Then, the test set sample was added to the next training set to extend the picture sample, and in the application of the algorithm, different types of forest fire image samples were added to the database. The forest fire image database can be gradually expanded to enhance the algorithm's recognition accuracy towards forest fire images.

It can be seen from Table 5 that the accuracy of this algorithm is higher than that of a convolutional neural network algorithm, improving the real-time performance of fire warning while ensuring accuracy. The time consumption of this algorithm is only 1/4 of the convolutional neural network algorithm. The deep-learning method has high requirements regarding equipment performance and has a long training time. The proposed algorithm greatly reduces the training and prediction time.

**Table 5.** Comparison of the proposed algorithm and deep-learning algorithms.

| Recognition Algorithm | The Proposed Algorithm | Convolutional Neural Network |
|---|---|---|
| Identification accuracy | 93.15% | 91.42% |
| Total time-consuming | 9.22 min | 28.30 min |
| Recognition rate | 0.42 s | 1.29 s |

## 4. Conclusions

This paper proposed a forest fire recognition method based on fusion color and textural features. The suspected fire region was segmented via the fusion RGB-YCbCr color space. Then, the textural features of the suspected fire region were extracted via LBP and GLCM algorithms to form a 14-dimensional textural feature vector. Finally, the forest fire image feature database was established, and the support vector machine was used for forest fire recognition and classification. The results show that the algorithm's accuracy of recognizing flames in forest fire images can reach 93.15%, and the algorithm has good robustness when fire-like interference appears. In the future, it is proposed to further improve the forest fire image database and expand the training set to include more forest fire images with different shapes, sizes, colors, and burning degrees; add more forest images with fire-like interference and classify them according to different features; and further reduce the algorithm's training and testing times to provide new concepts for the application of image recognition in forest fire prevention.

**Author Contributions:** Conceptualization and methodology, C.L.; software and investigation, Q.L.; validation and formal analysis, B.L.; data curation and writing—original draft preparation, L.L. All authors have read and agreed to the published version of the manuscript.

**Funding:** This work was supported by the National Natural Science Foundation of China (Grant No. U2033206), the National Key Research and Development (R&D) Plan (Grant No. 2018YFC0809500), Science and Technology Project of State Grid General Aviation Company Limited

(Grant No. 1100/2021-440038), Guizhou Scientific Support Project (Grant No. 2021 General 514), and the Project of the Excellent Youthful Teacher of Fundamental Research Funds for the Central Universities (Grant No. 2022YQAQ05).

**Conflicts of Interest:** The authors declare no conflict of interest.

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
