# Peer review of "Investigation of Recognition and Classification of Forest Fires Based on Fusion Color and Textural Features of Images"

_forests, doi:10.3390/f13101719_

Round 1
Reviewer 1 Report
It is advised to compare the method suggested with the existing recent works. Consider the case of False Positive and observe the behavior of the method in that case. Evaluation parameters should be specified separately in a subsection. Provide details and justification for the SVM kernel used. How the system behave for different choices of classification methods other than SVM?
Author Response
Dear Editor:
We would like to submit the revised manuscript entitled “Investigation of recognition and classification of forest fire based on fusion color and texture features of images”, which we wish to be considered for publication in “forests”.
We are very grateful to you for giving us the opportunity to transfer and also thanks much for the reviewers’ significant comments. Those comments are all valuable and very helpful for revising and improving our paper, as well as the important guiding significance to our researches. We have studied comments carefully and have made correction which we hope meet with approval. Here are our point-by-point responses:
Reviewer 1:
Comments and Suggestions for Authors
It is advised to compare the method suggested with the existing recent works. Consider the case of False Positive and observe the behavior of the method in that case. Evaluation parameters should be specified separately in a subsection. Provide details and justification for the SVM kernel used. How the system behave for different choices of classification methods other than SVM?
Reply: Firstly, thank you very much for your comments. We compare this algorithm with the convolutional neural network algorithm, and find that this algorithm has a smaller difference in recognition rate than deep learning, but it takes less time and requires lower performance of the equipment. Evaluation parameters have been specified separately in a subsection. We have provided details and justification for the SVM kernel used. We explain the content of the RFB kernel in detail, and explain why we choose the RFB kernel as the function kernel of the support vector machine. For the selection of different classification methods other than SVM, we have taken convolutional neural network as an example, used the data of this paper to identify, and compared the results. The revised version is as follows:
In this section, the support vector machine was used to construct a decision function that recognize and classify the forest image to the three scenes. The purpose of SVM is to construct a decision function that can classify data to the greatest extent. All sample data meet the following formula :
|
(12) |
Where is the normal vector of the hyperplane, b is the intercept of the hyperplane, and C is the penalty parameter. The LBP histogram distribution feature and GLCM feature of forest fire flame image and interference image are extracted from the existing samples to form 14-dimensional vector: X=[,,,…,,,,,]. The collected data are used to establish training set and test set, and Radial Basis Function kernel is used to identify and classify forest fire flames [36-38]. Radial Basis Function kernel is defined as follows:
|
(13) |
Where p is the width of the radial basis kernel function. Using Radial Basis Function kernel function to identify forest fires is effective. RBF can analyze high-dimensional function, the main identification classification steps are as follows
(1) Training process : The target image is extracted as the training set. Based on the LBP histogram feature of the target image and the GLCM texture extraction feature as the image feature input to the SVM vector machine for classifier training.
(2) Recognition process : The LBP and GLCM features of the recognition image are extracted and classified by the trained classifier. Finally, the classification performance of the classifier is evaluated by the accurate recognition results.
This paper is based on support vector machine to identify and classify forest fires, and combines comparative convolutional neural networks to verify the accuracy and effectiveness of the algorithm. Convolutional Neural Networks is a kind of Feedforward Neural Networks with convolution calculation and deep structure, which is one of the representative algorithms of deep learning. Convolutional neural network has a mature application in the field of computer vision, this paper uses its image recognition capability for forest fire image recognition. This algorithm first uses the fusion color space rule to extract the suspected flame area, then uses the LBP-GLCM method to extract the texture features, and finally inputs the texture features into the support vector machine. As a comparison algorithm, the article inputs the extracted image of the suspected flame area into the convolutional neural network.

Reviewer 2 Report
1. Is there any specific reason not to use CNN? This kind of segmentation method problem will be carried out with the use of a Convolutional Neural Network.
2. CNN must be incorporated at least in the comparative section.
3. Comparative study with other existing methods is a must.
5. An algorithm for the SVM-based model has to be written.
6. Discussion before a conclusion has to be added.
7. Cite the following papers:
a) Liu, J., Wang, X., & Wang, T. (2019). Classification of tree species and stock volume estimation in ground forest images using Deep Learning. Computers and Electronics in Agriculture, 166, 105012.
b) Roy, S. S., Goti, V., Sood, A., Roy, H., Gavrila, T., Floroian, D., ... & Mohammadi-Ivatloo, B. (2014). L2 regularized deep convolutional neural networks for fire detection. Journal of Intelligent & Fuzzy Systems, (Preprint), 1-12.
c) Seifert, E., Seifert, S., Vogt, H., Drew, D., Van Aardt, J., Kunneke, A., & Seifert, T. (2019). Influence of drone altitude, image overlap, and optical sensor resolution on multi-view reconstruction of forest images. Remote sensing, 11(10), 1252.
c) Balas, V. E., Roy, S. S., Sharma, D., & Samui, P. (Eds.). (2019). Handbook of deep learning applications (Vol. 136). New York: Springer.
d)Samui, P., Roy, S. S., & Balas, V. E. (Eds.). (2017). Handbook of neural computation. Academic Press.
Reviewer 3 Report
Topic of this manuscript is very relevant, climate change can cause more serious forest fires than we had before, therefore the research is very actual. The title covers the messages of the main body text. This manuscript contains the main elements scientific paper requires, Introduction is enough detailed, methods are acceptable. Mathematical formulas are correct. Results are nice and has a contribution.
Reviewer doubt is only that how the results can help practically to reduce the fire risk or seriousness. Even if this paper focuses on the research, and the introduction detailed the previous works and conclusion contains direction to further research, I suggest to the authors to give some more practical approach for the benefit to their results.
Author Response
Dear Editor:
We would like to submit the revised manuscript entitled “Investigation of recognition and classification of forest fire based on fusion color and texture features of images”, which we wish to be considered for publication in “forests”.
We are very grateful to you for giving us the opportunity to transfer and also thanks much for the reviewers’ significant comments. Those comments are all valuable and very helpful for revising and improving our paper, as well as the important guiding significance to our researches. We have studied comments carefully and have made correction which we hope meet with approval. Here are our point-by-point responses:
Reviewer 3:
Comments and Suggestions for Authors
Topic of this manuscript is very relevant, climate change can cause more serious forest fires than we had before, therefore the research is very actual. The title covers the messages of the main body text. This manuscript contains the main elements scientific paper requires, Introduction is enough detailed, methods are acceptable. Mathematical formulas are correct. Results are nice and has a contribution.
Reviewer doubt is only that how the results can help practically to reduce the fire risk or seriousness. Even if this paper focuses on the research, and the introduction detailed the previous works and conclusion contains direction to further research, I suggest to the authors to give some more practical approach for the benefit to their results.
Reply: This algorithm uses flame color and texture as extraction features, and is accompanied by RGB, YCBCR, LBP, GLCM and other flame feature extraction methods. At the same time, it combines the relevant knowledge of this major to integrate and improve the algorithm based on the actual forest fire situation. It can accurately and effectively identify forest fires of a certain scale, and can be effectively applied in fire monitoring and fire extinguishing operations and other operations involving forest fires, avoiding problems such as high cost of manual inspections and untimely detection. Extracting real-time forest images from hardware such as forest cameras, realizing the whole process of fire monitoring and identification and accurately dividing the flame area, timely and effectively sending out monitoring and early warning information and getting the actual location of forest fires at the first time, it also provides effective auxiliary decision-making assistance for subsequent fire extinguishing operations, avoiding errors and inefficiency caused by manual identification. At the same time, this paper focuses on extracting multiple features of the flame itself for comprehensive identification, and there are still few related studies on comprehensive identification. In addition, forest fires will cause great economic and property losses. At present, there is still a lack of systematic, holistic and multi-angle forest fire monitoring and recognition system, and the accuracy rate cannot be guaranteed.
In the revised manuscript, the practical approach for the benefit to the results has been added in the section of Introduction and Conclusions.

Reviewer 4 Report
This paper presents a classification methodology to identify forest fires in digital still images, using a combination of colour- and texture- analysis and support vector machines for classification. The methodology itself appears useful and generally applicable, but the paper suffers from poor description of methods, lack of context of the research (authors don't explicitly make clear the intended use of the method - eg. for classifying still images taken from digital cameras on the ground - it was some time into reading before I realized the method did not relate to satellite or motion video). The methods, as presented, assume a lot of reader knowledge of image analysis methods, and do not define acronyms of adequately describe the background of each element of the analysis. Given the readership of the Forests journal and this lack of detail, I wonder if the paper may be more suitable for a machine learning or image analysis journal.
The paper would also benefit from extensive English language editing, as the grammar is often confusing.
Detailed comments:
Line 22: Do the authors mean "forest fire area" when they say "forest area"?
Introductory paragraph could be expanded to make more explicit the comparisons - for example, make clear this is about classifying ground-level still photographs (as opposed to satellite imagery, another common fire detection method), make clear what the "traditional manual method" mentioned is.
The paper uses a lot of acronyms without defining them at all, for example, line 34 "ISSA", line 35 "FSCN", line 40 "HOPC-TOP", line 51 "UFS", and also uses a lot of technical phrases without defining them eg. line 40 "three-phase cross planes". This is a problem throughout the paper.
Line 53. "Wang et al." citation is written twice.
Line 57. Unsure what "excessive project caused by blind identification" means,
Line 59. Unsure what "vegetation is normalized" means.
Line 72: Do the authors mean "increases" the error rate, rather than "improves"?
The introduction paragraphs detailing past methodologies reads like a list of past studies, but lacks context - which methods work well in which circumstances? Which methods informed the development of this research, and which did not? Why is there need for a new method? Are these slow/inaccurate?
Line 92: "Ashour et al." citation is written twice.
Line 117 - Can the authors give, here, some examples of "fire-like interference sources"? More examples are given in the results, but the concept should be properly described here.
Line 122: LBP and GLCM are important acronyms and algorithms used throughout the paper - define them when first introduced.
Figure 1- Authors state above there are four main steps - it may be useful to indicate those four steps on this flowchart. Also, I don't believe the "correction algorithm" shown on this flowchart is described in the paper.
Line 145: What is the scale of these thresholds, eg. is it 0-100, or is it 0-255 (eg. 8-bit pixel values)?
Figure 2. What do the numbers under each image represent? Is it the "pixel retention rate" as mentioned in the method? Make clear in caption and explain what this represents.
Line 158: How was "manual extraction" performed?
Equation 2, line 175 - do the "RGB" columns (I assume) of this matrix require labels?
Line 177: I think C(rmean) here should be C(bmean)
Equations 5,6,7 - Define terms in the equations - this section assumes reader has understanding of LBP algorithm, which is unlikely given journal scope.
Line 229: What are the P,R parameters here? Why were 8,1 and 8,2 specifically chosen? Is there an a priori reason, or were tests of performance performed?
Line 280: define "RBF".
General question: The practical application of this methodology requires the placement of cameras in forests. Presumably, for rapid accurate detection of actual fires, you would require either a) many cameras, or b) cameras viewing a forest at a distance so they detect over a large area. How well does this algorithm perform with small, distant fires? What sort of detection area can it operate over? Do the texture elements differ between detecting close-up fire (as in many of the example images) and distant fire that may be small in the image field?
Distance
Author Response
Dear Editor:
We would like to submit the revised manuscript entitled “Investigation of recognition and classification of forest fire based on fusion color and texture features of images”, which we wish to be considered for publication in “forests”.
We are very grateful to you for giving us the opportunity to transfer and also thanks much for the reviewers’ significant comments. Those comments are all valuable and very helpful for revising and improving our paper, as well as the important guiding significance to our researches. We have studied comments carefully and have made correction which we hope meet with approval. Here are our point-by-point responses:
Reviewer 4:
Comments and Suggestions for Authors
This paper presents a classification methodology to identify forest fires in digital still images, using a combination of colour- and texture- analysis and support vector machines for classification. The methodology itself appears useful and generally applicable, but the paper suffers from poor description of methods, lack of context of the research (authors don't explicitly make clear the intended use of the method - eg. for classifying still images taken from digital cameras on the ground - it was some time into reading before I realized the method did not relate to satellite or motion video). The methods, as presented, assume a lot of reader knowledge of image analysis methods, and do not define acronyms of adequately describe the background of each element of the analysis. Given the readership of the Forests journal and this lack of detail, I wonder if the paper may be more suitable for a machine learning or image analysis journal.
Reply: Thank you very much for your comments. This paper mainly uses color and texture as extraction features, and uses RGB, Ycbcr, LBP, GLCM and other flame color texture extraction methods to identify flames from multiple recognition angles, combined with relevant professional knowledge. It has effective applications in the design of forest fire field operations such as fire monitoring and control, fire warning and prevention, and fire extinguishing operations. The real-time forest image is extracted to realize the whole process of fire monitoring and identification and the accurate division of the flame area. The monitoring and early warning information is sent out in time and effectively, and the actual location of the forest fire is obtained at the first time. It also provides effective auxiliary decision-making help for the subsequent fire extinguishing operation, and avoids the error and inefficiency caused by manual identification. In the updated version of this article, the acronyms are fully explained to ensure that readers understand the relevant content in the first time.
The paper would also benefit from extensive English language editing, as the grammar is often confusing.
Detailed comments:
Line 22: Do the authors mean "forest fire area" when they say "forest area"?
Reply: According to your suggestion, "forest fire area" is indeed more in line with the text than "forest area". In the revised version, "forest area" has been changed to "forest fire area". Through consulting relevant information, according to the global satellite data compiled by many institutions, compared with 2001, the forest area destroyed by the fire now increases by about 3 million hectares every year.
Introductory paragraph could be expanded to make more explicit the comparisons - for example, make clear this is about classifying ground-level still photographs (as opposed to satellite imagery, another common fire detection method), make clear what the "traditional manual method" mentioned is.
Reply: In the revised manuscript, we have made it clear what the "traditional manual method" sentence is. The revised contents are as follows:
“Forest fire recognition technology based on image features has significant advantages such as high timeliness and high recognition rate, which makes it has the ability to identify forest fire as soon as possible to prevent forest fire expanding in scale, and replaces the traditional artificial lookout and secondary image recognition method with high investment and low income. ”
The paper uses a lot of acronyms without defining them at all, for example, line 34 "ISSA", line 35 "FSCN", line 40 "HOPC-TOP", line 51 "UFS", and also uses a lot of technical phrases without defining them eg. line 40 "three-phase cross planes". This is a problem throughout the paper.
Reply: Thank you for your advice. It is necessary to define acronyms and technical phrases. In the revised manuscript, we have defined and explained the relevant contents. The unfinished sentences have been revised.
Line 53. "Wang et al." citation is written twice.
Reply: Thanks for your careful check very much. In the revised the “Wang et al.” has been corrected. And the manuscript has been double checked to avoid more typos.
Line 57. Unsure what "excessive project caused by blind identification" means,
Reply: In the revised manuscript, we have revised this part as follows:
“Wang et al. [7] convolutional neural network is often used for image feature learning. Combining it with image processing, on the one hand, it can effectively and specifically learn flame recognition and extract corresponding features; On the other hand, it has good performance and efficiency. ”
Line 59. Unsure what "vegetation is normalized" means.
Reply: In the revised manuscript, we have revised this part as follows:
“Jiang et al. [8] based on the method of infrared image and flame spectrum threshold analysis to obtain the feature vector, so as to quickly locate the fire location. Correspondingly, it can effectively eliminate the interference caused by various interferences in the forest and various noises in its own scene. ”
Line 72: Do the authors mean "increases" the error rate, rather than "improves"?
Reply: Thanks for your careful read very much. According to your suggestion, "increases" is indeed more in line with the text than "improves". In the revised version, "improves" has been changed to "increases". In order to avoid more word meaning errors, the manuscript has been checked again.
The introduction paragraphs detailing past methodologies reads like a list of past studies, but lacks context - which methods work well in which circumstances? Which methods informed the development of this research, and which did not? Why is there need for a new method? Are these slow/inaccurate?
Reply: Thank you for your careful reading and valuable suggestions. The introduction section introduces the past methods in detail in three paragraphs, with clear structure. The first paragraph talks about the research on forest fire from the perspective of image recognition; The second paragraph is about the study of forest fire recognition from the perspective of color and texture features; The third paragraph is about the role of image recognition technology in fire fighting and forest management. At the same time, we have revised this part in the revised manuscript according to your suggestions.
Line 92: "Ashour et al." citation is written twice.
Reply: Thanks for your careful check very much. In the revised the “Ashour et al.” has been corrected . And the manuscript has been double checked to avoid more typos.
Line 117 - Can the authors give, here, some examples of "fire-like interference sources"? More examples are given in the results, but the concept should be properly described here.
Reply: According to your suggestion, we have given appropriate examples to describe fire like interaction sources. The notes are as follows:
“They are red objects that interfere with fire image recognition in the forest, such as banners, maple leaves, bottles, etc.”
Line 122: LBP and GLCM are important acronyms and algorithms used throughout the paper - define them when first introduced.
Reply: Thank you for your advice. We have defined LBP and GLCM in the original text. The unfinished sentences have been revised. The revised contents are as follows:
“A 14-dimensional vector of the forest fire image was formed by LBP (Local Binary Patterns) and GLCM (Gray-level co-occurrence matrix) algorithm.”
Figure 1- Authors state above there are four main steps - it may be useful to indicate those four steps on this flowchart. Also, I don't believe the "correction algorithm" shown on this flowchart is described in the paper.
Line 145: What is the scale of these thresholds, eg. is it 0-100, or is it 0-255 (eg. 8-bit pixel values)?
Reply: The threshold scale we use is 0-255.
Figure 2. What do the numbers under each image represent? Is it the "pixel retention rate" as mentioned in the method? Make clear in caption and explain what this represents.
Reply: Figure 2 The ' pixel retention rate ' in the method is represented below each image. This has been explained in the revised manuscript.
Line 158: How was "manual extraction" performed?
Reply: Taking artificial vision as the setting standard, the target flame pixel contour is depicted by Photoshop, and the target flame pixel area is extracted, and the extraction accuracy is set to 1. Compared with the RGB space extraction results, the flame extraction accuracy of different thresholds is obtained, as shown in table 1.
Equation 2, line 175 - do the "RGB" columns (I assume) of this matrix require labels?
Reply: We carefully compared the conversion between RGB and YCBCR and have now corrected it and added the matrix ' RGB ' column to it.
Line 177: I think C(rmean) here should be C(bmean)
Reply: After confirmation, this has been corrected in the revised manuscript.
Equations 5,6,7 - Define terms in the equations - this section assumes reader has understanding of LBP algorithm, which is unlikely given journal scope.
Line 229: What are the P,R parameters here? Why were 8,1 and 8,2 specifically chosen? Is there an a priori reason, or were tests of performance performed?
Reply: P is the surrounding pixel point, and R is the corresponding domain radius. In the algorithm research, we found that with the increase of the surrounding pixels and the radius of the pixel field, the feature vector and the operation time have an increasing trend. When the surrounding pixel is 16, the feature vector and operation time increase significantly, while when the surrounding pixel is 8 and the pixel domain radius is 2, the increase of feature vector and operation time is not obvious. Therefore, in combination with the actual monitoring and recognition, considering the fast and efficient factors, we selected the surrounding pixel point 8, the pixel field radius 2.
Line 280: define "RBF".
Reply:This is fully explained in the revised manuscript.
“The collected data are used to establish training set and test set, and Radial Basis Func-tion (RBF) kernel is used to identify and classify forest fire flames [36-38]. RBF is defined as follows:
|
(13) |
Where p is the width of the radial basis kernel function. Using Radial Basis Function kernel function to identify forest fires is effective. RBF can analyze high-dimensional function, the main identification classification steps are as follows
( 1 ) Training process : The target image is extracted as the training set. Based on the LBP histogram feature of the target image and the GLCM texture extraction feature as the image feature input to the SVM vector machine for classifier training.
( 2 ) Recognition process : The LBP and GLCM features of the recognition image are extracted and classified by the trained classifier. Finally, the classification performance of the classifier is evaluated by the accurate recognition results.”
General question: The practical application of this methodology requires the placement of cameras in forests. Presumably, for rapid accurate detection of actual fires, you would require either a) many cameras, or b) cameras viewing a forest at a distance so they detect over a large area. How well does this algorithm perform with small, distant fires? What sort of detection area can it operate over? Do the texture elements differ between detecting close-up fire (as in many of the example images) and distant fire that may be small in the image field?
Reply: Small distant fire is mainly reflected in the fire is more secretive, flame effective extraction area is small, depending on the camera performance will appear to capture the fire is not clear, low resolution, less flame color texture features and other issues. For the flame color, and the surrounding environment ( trees and flowers ) will be quite different, even if the effective area is small but can effectively capture the target range ; for less texture features, we have trained a large number of target images, which cover fire images with unclear fire source images and less texture features. The training results based on vector machine classification show that the algorithm has high accuracy in dealing with small remote fires and is less affected by hardware such as cameras.
When training vector machine classification, forest images used include various common trees in China, such as birch, pine, cypress, etc. Fire-like disturbances include red garbage, red stripes and red-leaf forests. The target images are forest field shooting, and the final recognition accuracy is high. Therefore, this algorithm can be applied to most forest field shooting situations.
For the distant fire and the close-up image of the fire, the overall flame texture features will not change significantly. When the image is enlarged several times to the close-up image, due to the resolution and clarity of the original image, a small number of pixels will be missing or unrecognizable. However, from the overall recognition effect and algorithm training results, the difference will not cause significant changes in the flame texture feature recognition results. This is because the overall flame texture has not changed, and the algorithm pays more attention to the recognition and monitoring of the overall flame texture features.

Round 2
Reviewer 4 Report
I thank the authors for their revised manuscript, which with the additional method details provided is much improved - just providing that additional information on the acronyms and technical terms makes it much easier to read and understand, although I still feel some moderate English language editing would benefit the paper.
There was one comment I made that was not addressed in the response:
Figure 1- Authors state above there are four main steps - it may be useful to indicate those four steps on this flowchart. Also, I don't believe the "correction algorithm" shown on this flowchart is described in the paper.
However this is a minor issue, I just felt it would be useful to make the graphic more informative.
I also thank the authors for their detailed response to my question about the distance scale the method can operate at. This paper would be improved if that issue could also be briefly addressed in the discussion - some additional information on what distance from fire the model was trained at, and how confident you are the method can detect fires at a distance, because this is key to the utility of the technique.
Author Response
Dear Editor:
We would like to submit the revised manuscript entitled “Investigation of recognition and classification of forest fire based on fusion color and texture features of images”, which we wish to be considered for publication in “Forests”.
We are very grateful to you for giving us the opportunity to transfer and thanks much for the reviewers’ significant comments. Those comments are all valuable and very helpful for revising and improving our paper, as well as the important guiding significance to our researches. We have studied comments carefully and have made correction which we hope meet with approval. Here are our point-by-point responses:
Reviewer:
Comments and Suggestions for Authors
I thank the authors for their revised manuscript, which with the additional method details provided is much improved - just providing that additional information on the acronyms and technical terms makes it much easier to read and understand, although I still feel some moderate English language editing would benefit the paper.
There was one comment I made that was not addressed in the response:
Figure 1- Authors state above there are four main steps - it may be useful to indicate those four steps on this flowchart. Also, I don't believe the "correction algorithm" shown on this flowchart is described in the paper. However, this is a minor issue, I just felt it would be useful to make the graphic more informative.
Reply:Thanks for your careful check very much. We modified Figure 1 to indicate those four steps on this flowchart to make our steps more logical, so that readers can better understand our article. We use Expand database to summarize and represent Expand database and correction algorithm, and describe them in the article. The revised version is as follows:
Firstly, preprocessed images and the suspected flame area was extracted based on the fused RGB- YCbCr color spaces. Secondly, the LBP and GLCM algorithms were used to extract the texture features of the suspected flame area, and a 14-dimensional texture feature vector was formed. Thirdly, the database of texture features was established based on a large number of training images, including the normal forest images, forest fire images, and fire-like interference images. Fourthly, forest fire can be identified by judging the similarity with three types of images in the database via support vector machine. And the accuracy of the method was further improved by expanding the forest fire image database continuously.
By using SVM classifier, the recognition rate of forest fire can reach more than 90 %, and the fusion algorithm can reach 93.15%, which can accurately identify forest fire. Then add the test set sample to the next training set to extend the picture sample and in the application of the algorithm, different types of forest fire image samples are added to the database. Gradually expand the forest fire image database to enhance the recognition accuracy of forest fire images.
I also thank the authors for their detailed response to my question about the distance scale the method can operate at. This paper would be improved if that issue could also be briefly addressed in the discussion - some additional information on what distance from fire the model was trained at, and how confident you are the method can detect fires at a distance, because this is key to the utility of the technique.
Reply:Thank you for your careful reading and valuable suggestions. This algorithm is based on color and texture image features to identify the characteristics of the flame, mainly used in short-range fire source identification. This is due to the recognition of the target flame of the short-range fire source. Its flame texture features are obvious and easy to extract, and are less affected by hardware such as the camera. At this time, the pixel points and texture features caused by the camera are not enough to cause the overall flame recognition to be blocked.
The same is true when identifying distant fire sources. Although there are cases of amplification or relatively few feature points, it is not enough to affect the overall fire source identification. That is to say, the main influencing factor of this algorithm is whether it can extract the overall and accurate flame texture. When we carry out vector machine learning and training, most of the flame features can be identified more completely and accurately. Among the 1317 images identified, the recognition accuracy of the algorithm can reach 93.15 %, which covers the fire background of multiple scenes such as medium and short-range fire sources and remote fire sources. Therefore, this algorithm in the forest fire image recognition monitoring can have a good effect.
The revised version is as follows:
A total of 1317 forest images were collected from the field, including 513 forest images without fire, 420 forest images with fire, and 384 forest images with fire-like interference. The forest images included a variety of common trees in China, such as birch, pine, cypress, etc. The fire-like interference included the red garbage, red stripes, and red leaf forest, etc. Forest flame images mainly choose close-range shooting or high-definition images, because its color and texture features more obvious.
